# Mixture of Attention Variants for Modal Fusion in Multi-Modal Sentiment Analysis

Chao He [1,2], Xinghua Zhang [3], Dongqing Song [1], Yingshan Shen [2], Chengjie Mao [1], Huosheng Wen [4], Dingju Zhu [4,*] and Lihua Cai [2,4,*]

[1]  School of Computer Science, South China Normal University, Guangzhou 510631, China;
     chaohe@m.scnu.edu.cn (C.H.); 2019022679@m.scnu.edu.cn (D.S.); 20091223@m.scnu.edu.cn (C.M.)
[2]  Aberdeen Institute of Data Science and Artificial Intelligence, South China Normal University,
     Guangzhou 528225, China; shenyingshan@scnu.edu.cn
[3]  International United College, South China Normal University, Guangzhou 528225, China;
     zhangxinghua@m.scnu.edu.cn
[4]  School of Software, South China Normal University, Guangzhou 528225, China; 20200666@m.scnu.edu.cn
[*]  Correspondence: zhudingju@m.scnu.edu.cn (D.Z.); lee.cai@m.scnu.edu.cn (L.C.)

**Abstract:** With the popularization of better network access and the penetration of personal smartphones in today's world, the explosion of multi-modal data, particularly opinionated video messages, has created urgent demands and immense opportunities for Multi-Modal Sentiment Analysis (MSA). Deep learning with the attention mechanism has served as the foundation technique for most state-of-the-art MSA models due to its ability to learn complex inter- and intra-relationships among different modalities embedded in video messages, both temporally and spatially. However, modal fusion is still a major challenge due to the vast feature space created by the interactions among different data modalities. To address the modal fusion challenge, we propose an MSA algorithm based on deep learning and the attention mechanism, namely the Mixture of Attention Variants for Modal Fusion (MAVMF). The MAVMF algorithm includes a two-stage process: in stage one, self-attention is applied to effectively extract image and text features, and the dependency relationships in the context of video discourse are captured by a bidirectional gated recurrent neural module; in stage two, four multi-modal attention variants are leveraged to learn the emotional contributions of important features from different modalities. Our proposed approach is end-to-end and has been shown to achieve a superior performance to the state-of-the-art algorithms when tested with two largest public datasets, CMU-MOSI and CMU-MOSEI.

**Keywords:** multi-modality; attention mechanism; sentiment analysis; feature fusion; deep learning

## 1. Introduction

The accessibility of 5G networks and the popularity of social media have given rise to the ubiquity of opinionated video messages in our cyber world. Amateur users are producing large numbers of videos on social media platforms such as TikTok and on the internet to share their sentiments and emotions toward all aspects of their daily lives [1,2]. These multi-modal data consist of at least two modalities, commonly including texts, acoustics, and images [3]. For instance, videos on TikTok not only contain the creator's spoken language but also their body movements and facial expressions with pleasing background music and special animation effects.

The rich information contained in multi-modal data creates immense opportunities for different organizations. The ability to digest multi-modal data is beneficial for a variety of application scenarios that can greatly enhance user experiences and utility. For example, in autonomous driving, the vehicular control unit can leverage cameras to monitor drivers' emotions, driving behaviors, and fatigue conditions in real-time and provide necessary feedback based on the multi-modal signal [4]. This can effectively enhance driving safety

and reduce accidents. In the field of medical and health services, multi-modal information (e.g., patient counseling recordings) can be applied for emotion tendency assessments to assist doctors in their decisions on patient treatments [5]. Social media platforms can adopt machine learning techniques to automatically poll collective sentiment tendencies on videos of a given topic, which can then be relayed to other relevant organizations for decision making [6]. However, with greater possibilities come greater challenges with leveraging multi-modal data. Unlike single-modality data (e.g., texts) for sentiment analysis, the diverse modalities contained in multi-modal data both complement and interfere with each other, making information fusion extremely challenging.

The challenges with fusing multi-modal data mainly lie in the encoding both single-modality and cross-modality information and the modeling of contextual relationships among the targeted units of analysis (e.g., utterance). Both tasks require significant computing in an enormous feature space, which invalidates manual feature engineering as a solution option. Fortunately, deep learning has shown promising potential for modeling multi-modal data, particularly for Multi-Modal Sentiment Analysis (MSA) tasks [7]. Various deep learning methods, including CNN, LSTM/GRU, (Self-)Attention, and BERT, have been leveraged to learn encoding from single and cross modalities [7] and have been shown to achieve state-of-the-art performances in MSA tasks [7]. However, there is still no consensus on how to efficiently fuse multi-modal information to remove existing noise while taking contexts into considerations for optimizing the performance of MSA.

In this work, we propose a novel algorithm based on a mixture of attention variants for multi-modal fusion in MSA tasks. Our approach divides the multi-modal fusion problem into two stages: (1) we leverage the self-attention module to maximize the intra-modality information value, and we use the BiGRU module to maintain inter-utterance contexts within each modality; (2) after compressing the feature space using a fully connected module, the tensors resulting from each modality are fed into four different attention variants for multi-modal fusion. In both stages, we apply the attention mechanism to distinguish the contributions from each modality, assigning higher weights to useful features, while reducing irrelevant background interference.

The main contributions of the current work can be summarized as follows: (1) We present a comprehensive literature review on both single and Multi-Modal Sentiment Analysis. (2) A novel MSA method, namely the Mixture of Attention Variants for Modal Fusion (MAVMF), is proposed to solve the multi-modal fusion challenge. (3) Experimental data on the two largest benchmark public datasets show that our proposed MAVMF algorithm can effectively extract multi-modal information, and it is shown to demonstrate improvements when compared with other baseline methods.

In the remaining sections, we provide a comprehensive summary of the sentiment analysis and new developments in Section 2, introduce our problem definition and proposed algorithm in Section 3, provide the details of the experiments in Section 4, and present the results in Section 5, followed by a discussion in Section 6 and a conclusion in Section 7.

## 2. Related Work

Before we entered into the video age in social media, text data dominated the Sentiment Analysis (SA) sphere and were considered the default data modality in SA. However, research on SA using only text data can suffer from issues like having an "emotional gap" and "subjective perception" [3], leading to unsatisfactory results in SA tasks. Compared with text-only data, user-recorded videos convey emotional information through subscripts (e.g., texts), images and the acoustic signals embedded in them. The popularity of video data gave rise to MSA, in which various modalities are leveraged to corroborate each other and provide a better recognition performance [8]. Since the effective extraction of features within single modalities serves as the foundation and prerequisite for MSA tasks, it is necessary to use feature extractors to extract internal features within each modality. In addition, since the fusion of multi-modal features is critical to the success of MSA tasks,

effective fusion algorithms are at the core of MSA research. From these two perspectives, we cover both single- and multi-modality SA in this section.

### 2.1. Single-Modality Sentiment Analysis

2.1.1. Text Sentiment Analysis

When conducting a text sentiment analysis (TSA), we aim to discover the emotional attitudes expressed by the authors by analyzing the emotions within the text. Before the advent of text analysis technology, people had to manually read and analyze the emotions conveyed in text, resulting in a significant increase in their workload. In addition, manual classifications are prone to human error. Therefore, the use of automation technology to infer text sentiments can significantly improve the label efficiency in TSA.

TSA tasks can be categorized into word-level, sentence-level, and document-level inferences [9]. Document-level SA focuses on the overall emotional tendency, which is obtained by assigning different sentiment contributions to different sentences and aggregating the sentiment tendencies of all sentences in the document. Sentence-level SA focuses on each individual sentence in a document and studies the emotional polarity of the sentence based on the sentiment contributions of the words within the sentence. Word-level SA focuses on each word that appears in a sentence and directly determines its emotional polarity through sentiment lexicons.

All TSA methods can be simply divided into rule-based and machine-learning-based methods [10]. The rule-based approaches use predesigned rules, such as sentiment lexicons, to determine text sentiments. For example, sentiment lexicons may define the polarity scores of emotion words, and the overall sentiment polarity is determined by aggregating the positive and negative scores of the words. The one with the higher score is selected as the final sentiment polarity. The performance of rule-based SA methods largely depends on the accuracy of the scoring for each word and the comprehensiveness of the lexicon set. Due to its simplicity in implementation, rule-based SA methods are widely adopted by researchers and practitioners [11]. For instance, Thelwall et al. [12] proposed the SentiStrength algorithm; Saif et al. [13] developed the SentiCircles platform for SA on Twitter; Li et al. [14] constructed a lexicon to effectively enhance the sentiment analysis performance; Kanayama et al. [15] proposed a syntactic-based method for detecting the sentiment polarity; and Rao et al. [16] conducted a sentiment analysis based on the document topic classification.

The machine-learning-based SA methods aim to automate SA tasks by using supervised models. For example, Chen et al. [17] proposed a novel SA algorithm to extract sentiment features from mobile app reviews and used Support Vector Machines (SVMs) for sentiment classification. Zhao et al. [18] used supervised algorithms to perform a binary classification on product review data, classifying the comments into positive and negative categories. Kiritchenko et al. [19] proposed a SVM algorithm for short and informal texts. Silva et al. [20] applied ensemble learning by combining various classifiers such as random forests and SVMs. More recently, deep learning approaches have emerged as a new avenue of research in TSA. Kim et al. [21] used Convolutional Neural Networks (CNNs) for SA and was able to demonstrate an excellent performance. Makoto et al. [22] combined spatial pyramid pooling with max pooling and used gated neural networks to classify user review texts. Meng et al. [23] proposed a multi-layer CNN algorithm and was able to prove its superiority through experiments. Jiang et al. [24] combined long short-term memory (LSTM) networks [25] with CNNs to handle the dependency on distant sentences. Luo et al. [26] introduced a gated recurrent neural network (RNN) to enhance the contextual relationships between words and texts. Minh et al. [27] proposed three variants of neural networks [28] to capture the long-term dependencies of information.

2.1.2. Image Sentiment Analysis

Image Sentiment Analysis (ISA) mainly focuses on the modeling of users' facial expressions and postures contained in an image to infer their emotional tendencies. Colombo et al. [29] first proposed an automatic emotion retrieval system that effectively ex-

tracts image features and performs emotion classification. Singh et al. [30] applied the CNN with domain specific fine tuning to classify sentiments on Flickr images. Yang et al. [31] created a learning framework that explores only the affective regions in an image and combined it with a CNN to classify the sentiment for an image. Yang et al. [32] proposed a weakly supervised coupled CNN with two branches to leverage localized information from an image for ISA. Kumar et al. [33] constructed a visual emotion framework for emotion feature extraction using the Flickr dataset. Truong et al. [34] developed item-oriented and user-oriented CNNs to better capture the interaction of image features with the specific expressions of users or items for the inference of user review sentiments. You et al. [35] extracted features from local image regions and conducted an ISA by incorporating an attention mechanism into the proposed network. Wu et al. [36] proposed a scheme for ISA that leverages both the inference on the whole image and subimages that contain salient objects. Zheng et al. [37] introduced an "Emotion Region Attention" module, while Li et al. [38] proposed a novel SentiNet model for ISA.

2.1.3. Speech Emotion Analysis

Compared to text and image SA tasks, the development of Speech Emotion Analysis (SEA) has been relatively slow. SEA focuses on analyzing emotions based on factors such as the tone, bandwidth, pitch, and duration of user speech [39]. Since deep learning techniques have been shown to improve the speech recognition performance [40], researchers have proposed various neural-network-based models to enhance the accuracy of speech emotion recognition [41–43].

*2.2. Multi-Modal Sentiment Analysis*

Existing and emergent social media platforms have enabled common users to post self-recorded videos to share their day-to-day living experiences and sentiments on any subject. This led to an explosion of multi-modal information on the internet and created tremendous opportunities for MSA [44]. Morency et al. [1] created the YouTube dataset and constructed a joint model to extract multi-modal features for SA. Poria et al. [45] applied single-modality feature extractors (e.g., CNN on text embeddings and Part-of-Speech taggings) on the visual, audio, and textual channels and trained a multi-kernel learning classifier for MSA. Zadeh et al. [2] introduced a multi-modal lexicon to better capture the interactions between facial gestures and speech. They also published the CMU-MOSI [46] dataset, which became the first benchmark dataset to support research in MSA. Zadeh et al. [47] presented a tensor fusion network model, which learns the interactions within and between text, vision, and acoustic channels. Chen et al. [48] proposed a novel SA model that comprises a gated multi-modal embedding module for information fusion in noisy environments and an LSTM module with temporal attention for higher-resolution word-level fusion.

The aforementioned works only considered the fusion of information between modalities without considering the dependencies between contexts. In order to improve the performance of MSA, Poria et al. [49] introduced an LSTM-based framework that captures the mutual dependencies between utterances using contextual information. In another study, Poria et al. [50] proposed a user-opinion-based model that combines the three modality inputs using a multi-modal learning approach. Zadeh et al. [51] proposed multiple attention blocks to capture information from the three modalities.

More recently, Wang et al. [52] proposed a novel Text Enhanced Transformer Fusion Network (TETFN) method that learns text-oriented pairwise cross-modal mappings to obtain effective unified multi-modal representations. Yang et al. [53] applied BERT to translate visual and audio features into text features to enhance the quality of both visual and audio features. Wu et al. [54] extracted bi-modal features from the acoustic–visual, acoustic–textual, and visual–textual pairs with a multi-head attention module to improve video sentiment analysis tasks. Wang et al. [55] created a lightweight Hierarchical Cross-Modal Transformer with Modality Gating (HCT-MG) for MSA through primary modality

identification. He et al. [56] proposed the Multi-Modal Temporal Attention (MMTA) algorithm, which considers the temporal effects of all modalities on each uni-modal branch to balance the interactions between unimodal branches and the adaptive inter-modal balance. Mai et al. [57] leveraged the contrastive learning framework both within modalities and between modalities for the MSA tasks.

Although these scholars have achieved promising results, there is still room for improvement. In SA tasks, within-modality representations are as important as inter-modality information fusion. The aforementioned research methods do not fully address the extraction of modality-specific features or the fusion of information between modalities. Particularly, the interaction and fusion of multi-modal features may lead to the presence of redundant information in the target network, making it challenging to focus on important information. Therefore, it is critical to identify the contributions of features from different modalities to SA at each stage of the deep learning network, which is the goal of our proposed MAVMF algorithm.

## 3. Method

### 3.1. Problem Definition

In this work, videos are considered the source of multi-modal data for sentiment analysis. A video usually contains a series of consecutive image frames, and a user's emotional tendency can be different or related in consecutive frames. Because of this, the video is processed into video utterances, each of which contains the same emotional tendency of the user, as shown in Figure 1. We aim to perform a sentiment analysis on the utterance level in a given video.

Assuming a dataset contains $m$ videos, $D = [V_1, V_2, \ldots, V_m]$. For the $i$-th video, $V_i$, the video is composed of $n_i$ video segments or utterances, $V_i = [u_{i1}, u_{i2}, \ldots, u_{in_i}]$, where $u_{i1}$ denotes the first utterance in $V_i$, and $n_i$ denotes the total number of utterances in $V_i$. For each utterance $u_{ij}, 1 \leq i \leq m$, and $1 \leq j \leq n_i$, contains a feature vector $u_{ij} = [t_{ij}, v_{ij}, a_{ij}]$, representing three modalities, i.e., $t_{ij}$ is the text feature representation, $v_{ij}$ is the visual feature representation, and $a_{ij}$ is the audio feature representation.

Assuming there are $C$ classes of emotional categories for the user's video, the goal is to label the emotional category of each video utterance. In order to perform sentiment classification, the utterances in the video, except for $u_{ij}$, are considered to be the context of $u_{ij}$, and the accuracy and F-1 score are used as the evaluation metrics for the model.

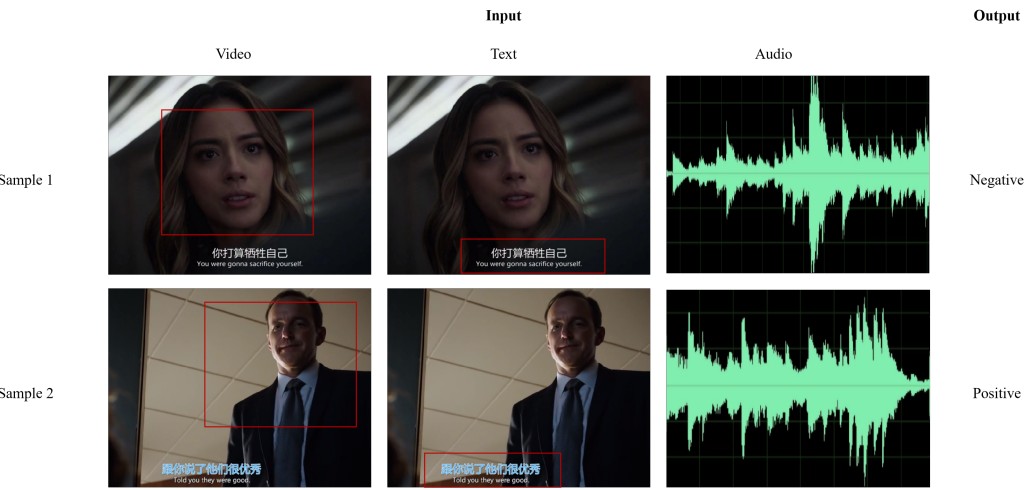

**Figure 1.** Selected samples from the CMU-MOSI dataset. Each sample represents an utterance from the given video. The output labels were obtained by using our proposed MAVMF algorithm.

### 3.2. Model

The MAVMF algorithm can be divided into five steps from an end-to-end pipeline as shown in Figure 2: (A) single-modal feature representation, (B) single-modal attention,

(C) contextual feature extraction, (D) multi-modal feature fusion, and (E) sentiment classification. The concrete architecture of MAVMF is illustrated in Algorithm 1. We employ ⊙ for the dot product, ⊗ for element-wise multiplication, and ⊕ for feature concatenation in all formulas in the below sections.

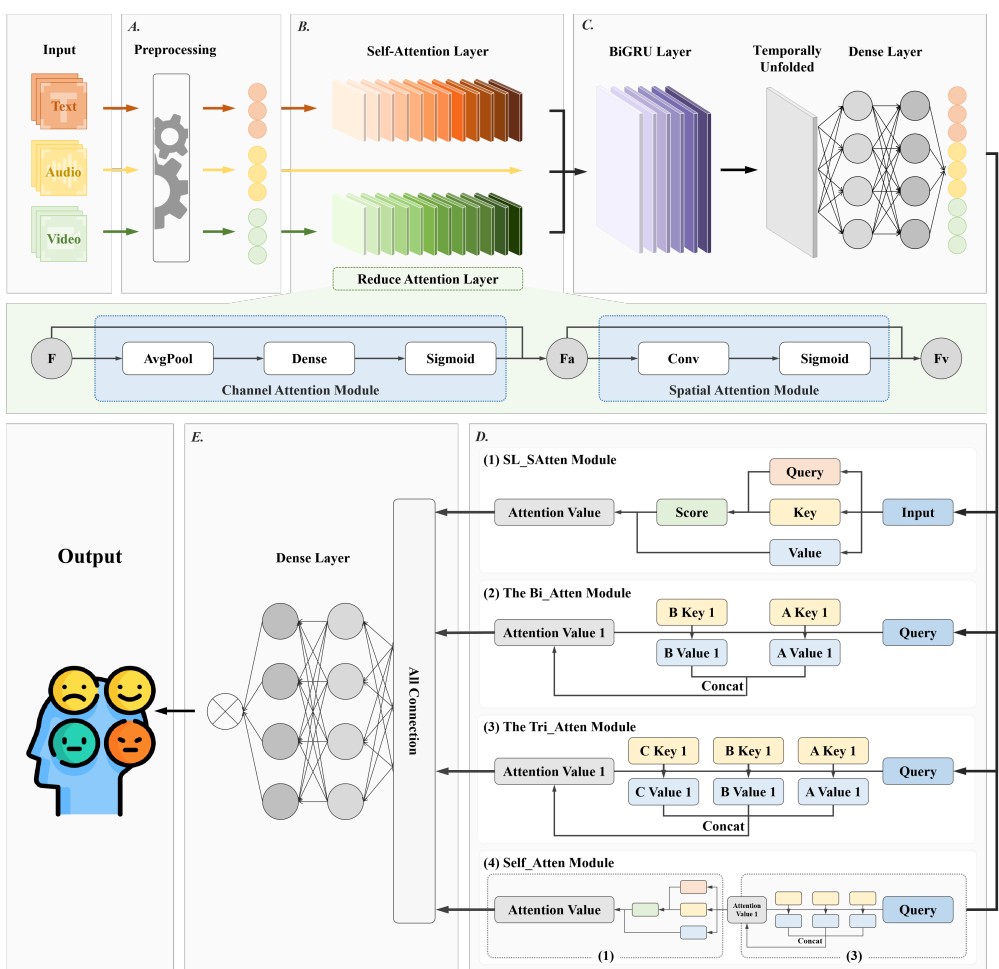

**Figure 2.** MAVMF architecture. (**A**) Preprocessing the input multi-modal data. (**B**) Single-modality self-attention and the reduced-attention module for the text and visual modalities. (**C**) BiGRU + fully connected dense layer(s). (**D**) Four attention and self-attention modules for modal fusion. (**E**) Concatenation + fully connected dense layer(s) with the softmax activation function for predictions.

### 3.2.1. Single-Modal Feature Representation

Due to the distinct semantic spaces of the text, image, and audio, different feature extractors should be used to extract the features within each modality. We adopt the following single-modal feature extractors that were chosen by other studies for the CMU-MOSI and CMU-MOSEI datasets. Specifically, for the CMU-MOSI dataset, we use the utterance-level features provided by Poria et al. [49] as inputs to the MAVMF model; for the CMU-MOSEI dataset, we employ the CMU-multi-modal data SDK [51] tool to extract the corresponding text, audio, and video features as inputs to the MAVMF model.

---

**Algorithm 1:** MAVMF architecture

---

**Input** : text *t*, audio *a*, visual *v* of train data *U* and test data *R*
**Output**: predictions for *R*
**Procedure TRAIN(*U*,*R*):**

   **Unimodal:**

      Train *BiGRU* and dense layers with *t*, *a*, and *v*
      $t \longleftarrow Self\_Atten(t)$
      $X_t \longleftarrow getBiGRUFeatures(t)$
      $D_t \longleftarrow Dense(X_t)$
      $v \longleftarrow RAtten(v)$
      $X_v \longleftarrow getBiGRUFeatures(v)$
      $D_v \longleftarrow Dense(X_v)$
      $X_a \longleftarrow getBiGRUFeatures(a)$
      $D_a \longleftarrow Dense(X_a)$

   **Multimodal:**

      Fusion of text, audio, and visual features
      $SL\_SAtten_t \longleftarrow SL\_SAtten(D_t)$
      $SL\_SAtten_v \longleftarrow SL\_SAtten(D_v)$
      $SL\_SAtten_a \longleftarrow SL\_SAtten(D_a)$
      $Bi\_Atten_{ta} \longleftarrow Bi\_Atten(concat(D_t, D_a))$
      $Bi\_Atten_{tv} \longleftarrow Bi\_Atten(concat(D_t, D_v))$
      $Bi\_Atten_{av} \longleftarrow Bi\_Atten(concat(D_a, D_v))$
      $Tri\_Atten_{tva} \longleftarrow Tri\_Atten(concat(D_t, D_v, D_a))$
      $Self\_Atten_{tva} \longleftarrow selfattention(Tri\_Atten_{tva})$

   **Fuse feature and classification:**

      $D \longleftarrow concat(D_t, D_v, D_a)$
      $O_{SL\_SAtten} \longleftarrow concat(SL\_SAtten_t, SL\_SAtten_v, SL\_SAtten_a)$
      $O_{Bi\_Atten} \longleftarrow concat(Bi\_Atten_{ta}, Bi\_Atten_{tv}, Bi\_Atten_{av})$
      $O_{out} \longleftarrow concat(D, O_{SL\_SAtten}, O_{Bi\_Atten},$
      $Tri\_Atten_{tva}, Self\_Atten_{tva})$
      $output \longleftarrow softmax(O_{out})$

   **Procedure TEST(*R*):**

      R is passed through the learned model to obtain the results
      $Y \longleftarrow MAVMF(t, a, v)$

---

### 3.2.2. Single-Modal Attention

The attention mechanism enables the target network to prioritize high-contributing features and disregard interference from the background information. This section primarily focuses on the single-modal attention modules, which employ distinct attention mechanisms to encode image and text features independently, enhancing the extraction of single-modal features. The single-modal attention module comprises two components: the reduced-attention (*RAtten*) block for the visual modality and the self-attention ($Self_Atten$) block for the text modality. The structure of the single-modal attention module is illustrated in Figure 2B.

**The RAtten Block**. The RAtten block employs channel attention and spatial attention to encode the internal features of the image, enhancing features that significantly contribute to emotions while suppressing background features. The image's feature vector sequentially passes through the channel attention and spatial attention modules, capturing the importance of each channel and feature map in the image. The specific process can be expressed as follows:

$$F_a = F \otimes \sigma(Dense(AvgPool(F))) \tag{1}$$

$$F_v = F_a \otimes \sigma(Conv(F_a)) \tag{2}$$

$F$ is the image information extracted by the preprocessing method described in Section 3.2.1, *AvgPool* is one-dimensional global average pooling, *Dense* represents the fully connected layer, $\sigma$ represents the *Sigmoid* function, $F_a$ represents the output of the image features $F$ after passing through the channel attention, *Conv* represents a one-dimensional convolution operation, and $F_v$ represents the output of the spatial attention.

**The Self_Atten Block**. The Self_Atten block applies the self-attention mechanism to encode text features, which can take into account the mutual influences among video segments. Since words in the same sentence in a video have different semantic associations, the self-attention mechanism can calculate the semantic associations between a word in a sentence and other words in the same sentence, providing them with different weights. This process can be expressed as follows:

$$m_i = x_i \odot x_i^T \tag{3}$$

$$n_i = softmax(m_i) \tag{4}$$

$$o_i = n_i \odot x_i \tag{5}$$

$$a_i = o_i \otimes x_i \tag{6}$$

$$t_i = x_i \oplus a_i \tag{7}$$

$x_i$ represents the text feature vectors extracted by the preprocessing method described in Section 3.2.1, $a_i$ is the output after self-attention and represents the importance of the different words in each utterance, $t_i$ is the output of the text features after passing through the Self_Atten block.

### 3.2.3. Contextual Feature Extraction

In order to capture the contexts and dependencies between utterances in each modality, we feed the single-modal attention features extracted in Section 3.2.2 for both the visual and text modalities and the preprocessed acoustic features to a *BiGRU* module separately. The *BiGRU* module consists of two Gated Recurrent Units (GRUs) with opposite directions, which can effectively capture the spatio-temporal information between video clip sequences and can also capture the forward and backward long-term dependencies between video clip sequences. The working principles of *BiGRU* can be expressed as follows:

$$r_t = \sigma(W_r x_t + U_r h_{t-1} + b_r) \tag{8}$$

$$z_t = \sigma(W_z x_t + U_z h_{t-1} + b_z) \tag{9}$$

$$\tilde{h}_t = tanh(r_t * U h_{t-1} + W x_t + b_h) \tag{10}$$

$$h_t = (1 - z_t) * h_{t-1} + z_t * h_t \tag{11}$$

$x_t$ is the input feature sequence of the current node, $h_{t-1}$ is the hidden layer state of the previous GRU unit of the current node; $r_t$ and $z_t$ are the reset gate and update gate of the GRU unit; $W_r$, $b_r$, $W_z$, $b_z$, $U_r$, and $U_z$ are the weight parameters of the target network; $\sigma$ is the corresponding *sigmoid* function of the network; and $*$ represents the multiplication of the corresponding feature vectors.

To better handle the heterogeneity within each modality and fully explore the internal correlations of single-modal features, the representation of each modality obtained from the above BiGRU module is unfolded temporally and fused through fully connected dense layer(s). This can be expressed as follows:

$$D_t = tanh(W_t B_t + b_t) \tag{12}$$

$$D_a = tanh(W_a B_a + b_a) \tag{13}$$

$$D_v = tanh(W_v B_v + b_v) \tag{14}$$

$B_t$, $B_a$, and $B_v$ are the output features of the text, audio, and video after going through the *BiGRU* module; $W_t$, $b_t$, $W_a$, $b_a$, $W_v$, and $b_v$ are the weight parameters of the target network; $D_t \in R^{u \times d}$, $D_a \in R^{u \times d}$, $D_v \in R^{u \times d}$ are the fully connected layer's text, acoustic, and visual information; $u$ represents the total number of sentences; and $d$ represents the number of neurons in the fully connected layer.

### 3.2.4. Multi-Modal Feature Fusion

This module comprises attention modules at four different levels and dimensions: (1) the sentence-level self-attention module; (2) the bi-modality attention module; (3) the tri-modality attention module; and (4) the self-attention module on (3). It is based on the outputs from Section 3.2.3.

(1) **SL_SAtten Module**. The contribution of internal features in each modality to users' emotional tendencies is often different. For example, in the sentence, "The weather is really good today! I really like this kind of weather", the word "like" contributes more to the users' emotions than the word "weather". Therefore, we propose a sentence-level self-attention mechanism, referred to as $SL - SAtten$, to select the emotional contributions of words within the modality at the sentence level. Taking the text modality as an example, we assume there are $u$ sentences in the text modality in total. For each sentence $x_i$, where $1 \le i \le u$, the working principle of the SL_SAtten module is as follows:

$$m_i = x_i \odot x_i^T \tag{15}$$

$$n_i = softmax(m_i) \tag{16}$$

$$o_i = n_i \odot x_i \tag{17}$$

$$a_i = o_i \otimes x_i \tag{18}$$

$$T = a_1 \oplus a_2 \ldots \oplus a_u \tag{19}$$

$$O_{SL-SAtten} = T \oplus V \oplus A \tag{20}$$

Here, $a_i$ is the output of the discourse $x_i$ after self-attention, indicating the importance of different words in each utterance. Then, we concatenate all of the outputs to obtain the corresponding text feature $T$. In the same vein, you can obtain the corresponding visual feature $V$ and the corresponding audio feature $A$. By concatenating the text, visual, and audio features, you can obtain the final output of the SL_SAtten module $O_{SL\_SAtten}$.

(2) **The Bi_Atten Module**. In order to improve the interactions between pairs of modalities in video data, a bi-modality attention module is proposed. This module aims to integrate two different modalities from different semantic spaces by enhancing the connections between them while eliminating the interference from background information in them, and thus the learning process will be able to focus on the associations between them. Taking text and visual modalities as an example, we suppose $D_t$ and $D_v$ are the output feature vectors of the text and visual modalities after going through the context feature extraction module, so its working principle is as follows:

$$m_1 = D_t \odot D_v^T \tag{21}$$

$$n_1 = softmax(m1) \tag{22}$$

$$o_1 = n_1 \odot D_v \tag{23}$$

$$a_1 = o_1 \otimes D_t \tag{24}$$

$$m_2 = D_v \odot D_t^T \tag{25}$$

$$n_2 = softmax(m_2) \tag{26}$$

$$o_2 = m_2 \odot D_t \tag{27}$$

$$a_2 = o_2 \otimes D_v \tag{28}$$

$$O_{BC-Atten}(vt) = a_1 \oplus a_2 \tag{29}$$

$O_{BC\_Atten}(vt)$ is the output feature vector fused from the video and text modalities, which can be used for subsequent emotion classification. In a similar way, we can obtain $O_{BC\_Atten}(av)$ for acoustic and visual modalities and $O_{BC\_Atten}(at)$ for acoustic and text modalities.

(3) **The Tri_Atten Module**. In order to model the interactions among all three modalities in a video, a tri-modal attention is proposed. Assuming the text, visual, and acoustic feature vectors after the context feature extraction module are $D_t$, $D_v$, and $D_a$, respectively, we first concatenate and fuse the text and image, text and acoustic, and image and acoustic modality information. Then, we use a fully connected network to map the information to the same semantic space, thereby initially fusing the information between different modality pairs. The process is shown in the following formulas:

$$F_{TV} = tanh((D_T \oplus D_V)W_{tv} + b_{tv}) \tag{30}$$

$$F_{TA} = tanh((D_T \oplus D_A)W_{ta} + b_{ta}) \tag{31}$$

$$F_{AV} = tanh((D_A \oplus D_V)W_{av} + b_{av}) \tag{32}$$

$W_{tv}$, $W_{ta}$, $W_{av}$, $b_{tv}$, and $b_{ta}$, $b_{av}$ are the weights and biases of the fully connected layer, and $F_{TV}, F_{TV}, F_{TV} \in R^{u \times d}$ represents the pairwise fused feature vectors, where $d$ is the number of neurons in the fully connected layer.

In order to further extract effective features, the feature vector of the third modality is multiplied by the results of the pairwise fused feature vectors obtained in Equations (30)–(32) to produce the matrix $C_k(k = 1, 2, 3)$. Then, the *softmax* function is used to calculate the attention distribution of the feature vector fusion results $P_k(k = 1, 2, 3)$, forming the tri-modal attention module $T_k(k = 1, 2, 3)$, and finally, we obtain the tri-modal fusion information $Tri_{ATV}, Tri_{VTA}, Tri_{TAV}$ through matrix multiplication operations. This is then concatenated to form a feature vector $O_{Tri\_Atten}$, which is the output of the Tri_Atten module. The process is shown in the following formulas:

$$C_1 = F_A \odot F_{TV}^T \tag{33}$$

$$C_2 = F_V \odot F_{TA}^T \tag{34}$$

$$C_3 = F_T \odot F_{AV}^T \tag{35}$$

$$P_1 = softmax(C_1) \tag{36}$$

$$P_2 = softmax(C_2) \tag{37}$$

$$P_3 = softmax(C_3) \tag{38}$$

$$T_1 = P_1 \odot F_A \tag{39}$$

$$T_2 = P_2 \odot F_V \tag{40}$$

$$T_3 = P_3 \odot F_T \tag{41}$$

$$Tri_{ATV} = T_1 \otimes F_{TV} \tag{42}$$

$$Tri_{VTA} = T_2 \otimes F_{TA} \tag{43}$$

$$Tri_{TAV} = T_3 \otimes F_{AV} \tag{44}$$

$$O_{Tri\_Atten} = Tri_{ATV} \oplus Tri_{VTA} \oplus Tri_{TAV} \tag{45}$$

$C_1, C_2, C_3 \in R^{u \times u}$, $T_1, T_2, T_3 \in R^{u \times d}$, and $O_{Tri\_Atten} \in R^{u \times 3d}$.

(4) **Self_Atten Module**. The output of the Tri_Atten module may carry redundant features. To filter out redundant information, we apply a self-attention module for feature selection. This process is shown in the following formulas:

$$m_i = O_{Tri\_Atten} \odot O_{Tri\_Atten}^T \tag{46}$$

$$n_i = softmax(m_i) \tag{47}$$

$$o_i = n_i \odot x_i \tag{48}$$

$$a_i = o_i \otimes x_i \tag{49}$$

$$O_{Self\_Atten} = x_i \oplus a_i \tag{50}$$

$O_{Self\_Atten}$ is the output feature vector of the Self_Atten module.

### 3.2.5. Multi-Modal Sentiment Classification

Finally, the multi-modal sentiment classification module concatenates and combines the feature vectors obtained above and uses a fully connected layer to integrate and classify sentiments based on both inter-modal and intra-modal information. It is shown as follows:

$$\begin{aligned} out = D_v \oplus D_t \oplus D_a \\ \oplus O_{SL-SAtten} \oplus O_{CS-SAtten} \\ \oplus O_{BC-Atten}(vt) \oplus O_{BC-Atten}(at) \\ \oplus O_{BC-Atten}(av) \oplus O_{Self-Atten} \end{aligned} \tag{51}$$

$$output = softmax(out) \tag{52}$$

*output* is the final output information for the MAVMF model.

## 4. Experiments

The experiments were conducted on a Windows system using an NVIDIA GeForce RTX 2060 graphics card with 8G running memory. Python was used as the programming language with the Keras framework. The effectiveness of the MAVMF model was validated on the two benchmark datasets, CMU-MOSI and CMU-MOSEI.

### 4.1. Data

(1) The CMU-MOSI dataset includes 93 videos sourced from YouTube, covering topics such as movies, products, and books. There are a total of 2199 utterances, each of which is labeled as positive or negative. In the experiment, we used training and test sets containing 62 and 31 videos, respectively.

(2) The CMU-MOSEI dataset includes 3229 videos with a total of 22,676 utterances, each with an emotional score in the range of $[-3, +3]$. For the purpose of sentiment classification, utterances with a score of greater than or equal to 0 were labeled as positive, while those with scores less than 0 were labeled as negative. In the experiment, we used training, test, and validation sets containing 2250, 679, and 300 videos, respectively. Detailed information about the CMU-MOSI and CMU-MOSEI datasets is shown in Table 1:

**Table 1.** The details of the CMU-MOSE and CMU-MOSEI datasets.

| Description | CMU-MOSI | | CMU-MOSEI | |
| --- | --- | --- | --- | --- |
| | Training Set | Test Set | Training Set | Test Set |
| # Video | 62 | 31 | 2250 | 679 |
| # Utterance | 1447 | 752 | 16216 | 4625 |
| # Pos Utterance | 709 | 467 | 11498 | 3281 |
| # Neg Utterance | 738 | 285 | 4718 | 1344 |

From the detailed information about the CMU-MOSE and CMU-MOSEI datasets presented in Table 1, it is apparent that the number of positive utterance samples in these two datasets is greater than the number of negative utterance samples, leading to an imbalance in the distribution of positive and negative samples. Therefore, the accuracy and F-1 scores are used as evaluation metrics for the models.

### 4.2. Parameter Tuning

During the experiments, we investigated the impacts of different learning rates and batch sizes on the model performance. The learning rates chosen were 0.05, 0.01, 0.005, and 0.001, and the batch sizes chosen were 32 and 64. The parameters that yielded the best results were used for the final model. The final parameter settings used for the MAVMF model are shown in Table 2:

**Table 2.** Experimental parameter settings.

| Parameter | Value |
| --- | --- |
| BiGRU unit | 300 |
| BiGRU dropout | 0.5 |
| fully connected unit | 100 |
| fully connected dropout | 0.5 |
| activation function | tanh |
| learning rate | 0.001 |
| batch processing | 32 |
| number of iterations | 64 |
| optimization function | Adam |
| loss function | categorical cross-entropy |

### 4.3. Baseline Models

To compare the performances of the MAVMF model in the MSA tasks, for the CMU-MOSI dataset, we used the following baseline methods:

(1)  GME-LSTM [48]: This model is composed of two modules. One is the gated multi-modal embedding module, which can perform information fusion in noisy environments. The other is an LSTM module with temporal attention, which can perform word-level fusion with a higher fusion resolution.

(2)  MARN [51]: This model captures the inter-relationships among text, images, and speech in a time series through multiple attention modules and stores the information in a long short-term hybrid memory.

(3)  TFN [47]: This model encodes intra-modal and inter-modal information by embedding subnetworks within a single modality and tensor fusion strategy.

(4)  MFRN [58]: This model first stores the modality information through a long short-term fusion memory network and fully considers the information of other modalities when encoding a certain modality, thereby enhancing the modality interactivity. Then, it further considers the information of other modalities when encoding a single modality through a modality fusion recurrent network. Finally, further information fusion is achieved through the attention mechanism.

(5)  Multilogue-Net [59]: Based on a recurrent neural network, this model captures the context of utterances and the relevance of the current speaker and listener in the utterance through multi-modal information.

(6)  DialogueRNN [60]: This model tracks the states of independent parties throughout the dialogue process and processes the information through a global GRU, party GRU, and emotion GRU units and uses it for emotion classification.

(7)  AMF-BiGRU [61]: This model first extracts the connections between contexts in each modality through the BiGRU, merges information through cross-modal attention, and finally uses multi-modal attention to select contributions from the merged information.

For the CMU-MOSEI dataset, we have the following baseline methods:

(1)   MFRN [58]: As described above.

(2)   Graph-MFN [62]: This model's concept is similar to that of the MFN model, except that Graph-MFN uses a dynamic fusion graph to replace the fusion block in the MFN model.

(3)   CIM-Att [63]: This model first uses the BiGRU to extract the intra-modal context features, inputs these context features into the CIM attention module to capture the associations between pairwise modalities, and then concatenates the context features and CIM module features for sentiment classification.

(4)   AMF-BiGRU [61]: As described above.

(5)   MAM [64]: This model first uses the CNN and BiGRU to extract features from the text, speech, and image signals and then applies cross-modal attention and self-attention for information fusion and contribution selection.

## 5. Results

### 5.1. CMU-MOSI

Table 3 presents a comparison of the experiments between the MAVMF model and the chosen baseline models on the CMU-MOSI dataset. The MAVMF model shows some improvements in both the classification accuracy and F-1 score. Specifically, the accuracy of the MAVMF model is increased by 5.81%, 5.21%, 5.21%, 4.21%, 1.12%, and 2.51%, 0.26% when compared to the GME-LSTM, MARN, TFN, MFRN, Multilogue-Net, DialogueRNN, and AMF-BiGRU models, respectively. The F-1 score of the MAVMF model is increased by 8.8%, 5.2%, 4.3%, 4.3%, 2.1%, 2.36%, and 0.18% when compared to the GME-LSTM, MARN, TFN, MFRN, Multilogue-Net, DialogueRNN, and AMF-BiGRU models, respectively.

**Table 3.** Comparison of the performance with different models.

| Network Model | CMU-MOSI | |
|---|---|---|
| | Accuracy (%) | F-1 |
| GME-LSTM [48] | 76.50 | 73.40 |
| MARN [51] | 77.10 | 77.00 |
| TFN [47] | 77.10 | 77.90 |
| MFRN  [58] | 78.10 | 77.90 |
| Multilogue-Net [59] | 81.19 | 80.10 |
| DialogueRNN [60] | 79.80 | 79.48 |
| AMF-BiGRU [61] | 82.05 | 82.02 |
| **MAVMF** | **82.31** | **82.20** |

### 5.2. CMU-MOSEI

Table 4 presents a comparison of the experiments conducted between the MAVMF model and the chosen baseline models on the CMU-MOSEI dataset. The MAVMF model shows some improvements in both the classification accuracy and F-1 score. Specifically, the accuracy of the MAVMF model is increased by 3.2%, 4.2%, 1.3%, 2.26%, and 0.1% when compared to the MFRN, Graph-MFN, CIM-Att, AMF-BiGRU, and MAM models, respectively. The F-1 score of the MAVMF model is increased by 2.08%, 2.48%, 1.88%, 1.3%, and 0.58% when compared to the MFRN, Graph-MFN, CIM-Att, AMF-BiGRU, and MAM models, respectively.

**Table 4.** Comparison of the performance on different models.

| Network Model | CMU-MOSEI | |
|:---:|:---:|:---:|
| | Accuracy (%) | F-1 |
| MFRN [58] | 77.90 | 77.40 |
| Graph-MFN [62] | 76.90 | 77.00 |
| CIM-Att [63] | 79.80 | 77.60 |
| AMF-BiGRU [61] | 78.48 | 78.18 |
| MAM [64] | 81.00 | 78.90 |
| **MAVMF** | **81.10** | **79.48** |

*5.3. Modality Analysis*

In order to further analyze the impacts of features from different modalities on the classification performance of the MAVMF model, experiments were conducted on the CMU-MOSI dataset for both bi-modal and tri-modal feature sets. The experimental results are shown in Figure 3. We use T, V, and A to represent the text, visual, and acoustic modalities, respectively.

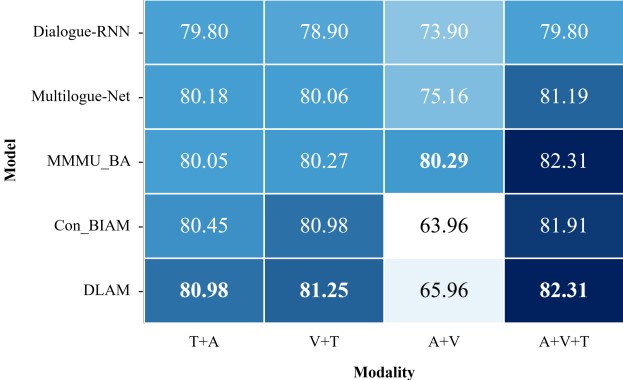

**Figure 3.** Visualization of the impacts of the modalities on the performances of the baseline algorithms and MAVMF.

From the above results, when compared to the selected baseline models, the classification accuracy of the text plus acoustic modalities is increased by 0.53–1.09%, the classification accuracy of the text plus visual modalities is increased by 0.27–2.35%, and the classification accuracy of the acoustic plus visual plus text modalities is increased by 0.4–2.51%. Apart from the fusion results for the video and acoustic modalities, the MAVMF model achieves the best performance for all other modality fusion methods. The fusion result for the acoustic and visual features is the worst, reflecting that the emotional expression polarity of the acoustic and visual modalities is weaker than that of text and that these modalities may be affected by background noise. This is consistent with the experimental results presented in the literature [65]. In addition, in MSA tasks, the classification performance obtained by fusing all three modalities is the best. This proves the necessity of leveraging multi-modal information in SA.

*5.4. Ablation Study*

To understand the impacts of the different modules applied in the MAVMF model, we conducted experiments on variants of the MAVMF model using the CMU-MOSI dataset and analyzed the experimental results. We included the following MAVMF variant models:

(1) MAVMF_Concat: This includes Modules A and E, as shown in Figure 2. (2) MAVMF_SAtten: This includes Module A, the self-attention module in Module B for the text modality, and Module E, as shown in Figure 2. (3) MAVMF_RAtten: This includes Module A, the reduced-attention module in Module B for the visual modality, and Module E, as shown in Figure 2. (4) MAVMF_RAtten_SAtten: This includes Module A,

Module B, and Module E, as shown in Figure 2. (5) MAVMF_BiGRU: This includes Module A, Module B, Module C, and Module E, as shown in Figure 2. (6) MAVMF_SL-SAtten: This includes Module A, Module B, and Module C, a sentence level self-attention module, and Module E, as shown in Figure 2. (7) MAVMF_Bi-Atten: This includes Module A, Module B, and Module C, a sentence level self-attention module, a dual modality cross modal attention module, and Module E, as shown in Figure 2. (8) MAVMF_Tri-Atten : This includes Module A, Module B, Module C, a sentence level self-attention module, a dual-modality cross-modal attention module, a tri-modality cross-modal attention module, and Module E, as shown in Figure 2. (9) MAVMF_Self-Atten: This includes Module A, Module B, Module C, Module D, and Module E, as shown in Figure 2.

Table 5 compares the proposed MAVMF model with its variant on the CMU-MOSI dataset. From the experiments, we see that the multi-modal sentiment classification accuracy of the MAVMF model gradually improves after adding each module. The text self-attention module, visual reduced-attention module, single-modality attention module, bidirectional gated recurrent unit module, sentence-level self-attention module, dual-modality cross-modal attention module, tri-modality cross-modal attention module, and self-attention module contribute 2.27%, 2.53%, 4.26%, 5.32%, 0.4%, 0.4%, 0.26%, 0.4%, and 0.53% to the classification accuracy, respectively. The marginal improvements become smaller as the complexity of the model increases.

**Table 5.** Comparison of the performance on different models.

| Network Model | CMU-MOSI | |
|---|---|---|
| | Accuracy (%) | F-1 |
| MAVMF_Concat | 70.74 | 71.01 |
| MAVMF_SAtten | 73.01 | 73.05 |
| MAVMF_RAtten | 73.27 | 73.24 |
| MAVMF_RAtten_SAtten | 75.00 | 74.95 |
| MAVMF_BiGRU | 80.32 | 80.21 |
| MAVMF_SL-SAtten | 80.72 | 81.09 |
| MAVMF_Bi-Atten | 81.12 | 81.10 |
| MAVMF_Tri-Atten | 81.38 | 81.41 |
| MAVMF_Self-Atten | 81.78 | 82.06 |
| MAVMF | **82.31** | **82.20** |

## 6. Discussion

Multi-Modal Sentiment Analysis tasks are commonplace in a diverse array of application scenarios. Video-based social media platforms, in particular, have empowered general users to generate an unprecedented amount of multi-modal data such as texts, audios, images, and the various combinations of them, which have enabled developers and practitioners to create multi-modal artificial intelligence systems that have already transformed our lives and work, as has been witnessed in the current wave of generative AI applications.

The work in this paper provides new insights into the fusion of multi-modal data for more general tasks beyond sentiment analysis. Our proposed MAVMF algorithm systematically explore the vast feature spaces that are generated by different modalities and their inherent spatial and temporal relationships. Unfortunately, when compared with other AI tasks such as image recognition and machine translation, the task of fusing multi-modal information for sentiment analysis remains an unsolved problem.

Currently, the underpinning theory regarding how different text, audio, and visual modalities complement or interfere with each other has not been formalized. However, one future direction in terms of solving the multi-modal fusion challenge could be relying on computation power, similar to what we have observed in large language models (LLMs). Doing this without leveraging a larger network with higher computation power may pose limits on the current performance of our proposed method. In other words, we did not consider constraints on the algorithm speed given a limited computation capacity, as the

full MAVMF model combines features from all proposed modules shown in Figure 2. However, we believe that, as computation is becoming cheaper and more accessible over time, there should be a focus on network design and modal fusion that encompasses all possible interactions among different modalities.

In our next step, we plan to investigate the performance of large foundation models in MSA tasks. Given that the current state-of-the-art performance in the existing literature on MSA tasks is still in the 80% range, we will first focus on improving the accuracy and robustness of novel algorithms for MSA tasks. Through pretraining with large amounts of unlabeled text and image data, foundation models have implicitly encoded large amounts of human knowledge in their weight parameters. It may be possible to adopt them when addressing MSA tasks. Certainly, we can also explore knowledge distillation techniques on successful foundation-model-based algorithms to obtain compact apprentice models for more resource constraint scenarios in MSA tasks.

## 7. Conclusions

To address the effective extraction of single-modal features and the efficient integration of multi-modal information, we propose an MSA algorithm, MAVMF. First, feature extractors are used to capture single-modal information. Then, for single-modal features, a reduced-attention module is used to encode the image, while a self-attention module is used for text. Subsequently, a bidirectional GRU and a fully connected network are applied to extract context-aware discourse features, capturing the context information between discourses in each modality. A sentence-level self-attention module is then used to model different types of modality information. At the modality level, dual-modality and tri-modality attention modules are applied to merge information, and a self-attention module is used to select features with significant contributions to reveal sentiment tendencies. Experiments on public datasets prove that, when compared to other deep learning algorithms, the MAVMF model has a better or comparable classification performance.

**Author Contributions:** Conceptualization, C.H., L.C. and D.Z.; methodology, C.H. and D.S.; software, C.H. and D.S.; validation, C.H. and D.S.; formal analysis, C.H. and D.S.; investigation, C.H.; resources, L.C., D.Z., X.Z., Y.S., C.M. and H.W.; data curation, C.H.; writing—original draft preparation, C.H., D.S., L.C., X.Z., Y.S., C.M. and H.W.; writing—review and editing, L.C., D.Z. and X.Z.; visualization, C.H., D.S. and X.Z.; supervision, L.C. and D.Z.; project administration, L.C. and D.Z.; funding acquisition, D.Z. Special thanks to Xinghua Zhang for valuable contributions to the graphical work. All authors have read and agreed to the published version of the manuscript.

**Funding:** This research received no external funding. The APC was funded by Prof. Dingju Zhu.

**Institutional Review Board Statement:** Not applicable.

**Informed Consent Statement:** Not applicable.

**Data Availability Statement:** Both the datasets applied in the current research are open datasets and can be found at the given link: http://multicomp.cs.cmu.edu/resources/ accessed on 1 January 2020. The datasets are available for downloading through the CMU Multimodal Data SDK GitHub: https://github.com/CMU-MultiComp-Lab/CMU-MultimodalSDK accessed on 1 January 2020.

**Conflicts of Interest:** The authors declare no conflicts of interest.

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
