# Peer review of "Mixture of Attention Variants for Modal Fusion in Multi-Modal Sentiment Analysis"

_2504-2289, doi:10.3390/bdcc8020014_

Round 1

Reviewer 1 Report

Comments and Suggestions for Authors

Comments on the Quality of English Language

 Moderate editing of English language required

Reviewer 2 Report

Comments and Suggestions for Authors

1. What is the main question addressed by the research?

The main question addressed by the research is how to leverage deep learning with attention mechanisms for multimodal sentiment classification.
The study aims to explore the effectiveness of the proposed approach in comparison to existing methods, particularly concerning its application to two major public datasets, CMU-MOSI and CMU-MOSEI.

2. Do you consider the topic original or relevant in the field? Does it address a specific gap in the field?

While the topic is relevant given the growing interest in multimodal sentiment analysis, the paper could benefit from a more explicit articulation of the novelty. A clearer identification of the specific gap in the existing literature that the proposed approach addresses would strengthen the paper.

3. What does it add to the subject area compared with other published material?

The paper contributes by presenting a comprehensive evaluation of a multimodal sentiment analysis method. However, a more thorough comparison with existing literature is needed to highlight the unique contributions and advancements brought by the proposed approach.

4. What specific improvements should the authors consider regarding the methodology? What further controls should be considered?

The authors should consider providing more details on the methodology to enhance clarity and reproducibility.

5. Are the conclusions consistent with the evidence and arguments presented, and do they address the main question posed?

The conclusions are generally in line with the evidence presented. Emphasizing how the results contribute to answering the main question would strengthen the overall coherence.
The authors could give some future plans.

6. Are the references appropriate?

The references provided are generally relevant.

7. Please include any additional comments on the tables and figures.

The authors provide tables for performance results of the models. However, the paper could benefit from some relevant figures.

ADDITIONAL COMMENTS:

Firstly, I appreciate the effort you have put into proposing a multi-modal sentiment analysis method based on deep learning and attention mechanisms. The comprehensive experiments conducted on the CMU-MOSI and CMU-MOSEI datasets demonstrate a rigorous evaluation of your approach against other algorithms.

However, there are concerns regarding the novelty of the proposed approach and the observed performance improvements over existing models. It appears that the novelty of the approach may not be clearly articulated, and the performance gains over other models seem marginal.

To strengthen your paper, consider addressing the following points:

1) Clearly articulate the novelty: Provide a more explicit explanation of the novel aspects of your approach. What specific contributions does your method bring to the field of multimodal sentiment analysis that distinguishes it from existing approaches?

2) Justify performance improvements: Elaborate on the significance of the observed performance improvements. Are there specific scenarios or aspects where your method outperforms others, and how does this contribute to the broader field of sentiment analysis?

I hope these comments help the authors.

Round 2

Reviewer 1 Report

Comments and Suggestions for Authors

The author has made good revisions, the article can be accepted after the following two points are noted. 

(1) Figure 3 is a bit blurry, please consider replacing it with a clearer one. 

(2) A blank character may be required between the serial number of the reference and the corresponding word. 

Reviewer 2 Report

Comments and Suggestions for Authors

Thank you for revising the manuscript and for your responses. The authors have satisfactorily addressed most of my concerns. In particular, the authors have greatly give explicit explanations.
